# TE-VLM: Transfer Entropy for Vision Language Model Distillation

## Abstract

Vision-Language Models (VLMs) have demonstrated impressive performance across various multimodal tasks. However, deploying large teacher models in real-world applications is often infeasible due to their high computational cost. To address this, knowledge distillation has been widely explored to transfer knowledge from a large teacher model to a smaller student model. In this paper, we propose a novel distillation framework that integrates Transfer Entropy (TE) as a regularization term to enhance information flow from the teacher to the student model. TE quantifies the directional dependency between teacher and student embeddings, encouraging the student model to effectively capture structural knowledge from the teacher. To efficiently approximate TE in high-dimensional embedding spaces, we introduce two surrogate formulations based on cosine similarity: (1) TE via cosine similarity of directional changes in embeddings and (2) TE via concatenated differences across modalities. Our experiments, conducted on the MSCOCO 2014 and Flickr8k datasets using CLIP-based teacher and student architectures, demonstrate that incorporating TE significantly improves retrieval performance. Through extensive analysis, we show that TE-based regularization enhances the student model's ability to capture multimodal associations and maintain representational consistency. Our findings suggest that TE is an effective tool for improving knowledge transfer in VLM distillation, bridging the performance gap between compact student models and their larger teacher counterparts.

## 1 Introduction

Vision-Language Models (VLMs) have emerged as a powerful framework for learning joint representations of images and text, enabling applications such as image captioning, visual question answering, and cross-modal retrieval (Radford et al., 2021; Jia et al., 2021). However, state-of-the-art VLMs are often computationally expensive, making them impractical for deployment in resource-constrained environments. To address this challenge, knowledge distillation (Hinton et al., 2015) has been widely adopted to transfer knowledge from a large teacher model to a smaller, more efficient student model while maintaining performance.

Existing approaches to VLM distillation primarily rely on contrastive learning (Li et al., 2022; Yang et al., 2024) and divergence-based losses, such as Kullback-Leibler (KL) divergence (Li et al., 2024b), to align the student model's probability distribution with that of the teacher. However, these methods do not explicitly quantify the directed information flow between the teacher and student representations. As a result, traditional distillation losses may overlook the sequential and structural dependencies present in the learning dynamics of multimodal embeddings.

To overcome these limitations, we propose a novel distillation framework that integrates Transfer Entropy (TE) as a regularization mechanism to enhance the student model's ability to mimic the teacher's information transfer patterns. TE is a measure of directed information flow between two systems, originally developed in information theory (Schreiber, 2000). In the context of VLM distillation, TE quantifies how much additional knowledge the teacher provides to the student beyond what the student has already learned from past states. This allows us to explicitly encourage meaningful knowledge transfer, ensuring that the student model captures the evolving structure of the teacher's representations.

The key contributions of this work are as follows:

- We introduce TE as a regularization method for VLM distillation, explicitly capturing the directed information flow from the teacher model to the student.

- We theoretically show that the first-order (linear) expansion of TE leads to a computable surrogate based on a cosine similarity between the teacher and student-process Jacobians.

- We propose two novel TE approximations based on cosine similarity, enabling efficient computation of information transfer in high-dimensional multimodal embeddings.

- We demonstrate that integrating TE into the distillation loss function leads to significant improvements in retrieval performance, outperforming traditional contrastive, KL-divergence, Mean Squared Error (MSE), and Interactive Contrastive Learning (ICL) distillation methods.

- We provide extensive empirical validation on the MSCOCO and Flickr8k datasets using different teacher-student distillation setups, showing that TE-based regularization enhances multimodal representation learning and improves student model alignment with the teacher.

## 2 RELATED WORK

### 2.1 KNOWLEDGE DISTILLATION

Knowledge distillation enables the transfer of learned representations from a large teacher network to a smaller student model (Hinton et al., 2015). Building on this idea, techniques utilizing intermediate representations have been developed to guide the training of deeper yet more efficient networks (Romero et al., 2015). Other approaches have leveraged attention mechanisms by emphasizing spatial attention maps (Zagoruyko & Komodakis, 2017) or addressed scenarios where original training data is unavailable through data-free methods (Huang & Wang, 2017).

Further research has focused on aligning internal representations between teacher and student networks. Contrastive methods harmonize feature spaces (Tian et al., 2019), while attention-based strategies have been tailored for transformer architectures (Touvron et al., 2021). Information-theoretic approaches have also emerged, either by maximizing mutual information (Ahn et al., 2019) or by capturing inter-sample relationships (Park et al., 2019). In addition, leveraging the probability distribution of the teacher network has proven effective in guiding the student (Passalis & Tefas, 2018). In (Liu et al., 2022), mutual relation distillation was proposed as a face recognition distillation method called CoupleFace. In (Chen et al., 2023), an objective function in multimodal representation learning was proposed to preserve the mutual information between the teacher and the auxiliary modality model for knowledge distillation.

Recent studies have ventured into the frequency domain. Frequency attention modules enable students to adjust feature representations under teacher guidance (Pham et al., 2024), and semantic frequency prompts have been employed to enhance dense prediction tasks (Zhang et al., 2024). Moreover, methods optimizing frequency representations have been proposed to generate compact synthetic datasets (Shin et al., 2023).

Other contributions in the area include self-distillation techniques for generating versatile text embeddings (Chen et al., 2024), strategies that synthesize minimal training samples to reduce computational overhead while preserving accuracy (Liu et al., 2024), dual-teacher frameworks (Li et al., 2024c), and the use of orthogonal projections to bolster knowledge transfer (Miles et al., 2024). An additional framework has been introduced to search for optimal distillation strategies tailored for object detection tasks (Li et al., 2024a).

### 2.2 VISION-LANGUAGE MODEL DISTILLATION

In the vision-language domain, early work aligned object semantics with textual descriptions to improve model performance (Li et al., 2020), while large-scale pre-training methods have been employed to learn universal image-text representations (Chen et al., 2020). Techniques adapting image-based models to video data have been proposed by leveraging high-quality pseudo-captions (Zhao et al., 2024), and methods to condense large datasets into smaller, information-rich synthetic sets have also been developed (Wu et al., 2023b).

Subsequent efforts have focused on enhancing reasoning and retrieval capabilities. Instruction-tuning frameworks have been devised to enable models to solve complex visual tasks through distilled reasoning abilities (Hu et al., 2024), and methods for open-vocabulary object detection via multimodal knowledge transfer have been explored (Gu et al., 2021). Approaches targeting video-language retrieval tasks (Pei et al., 2023) and incorporating frequency information to boost out-of-distribution generalizability (Li et al., 2023) further extend these ideas. Complementary techniques include methods leveraging vision-language models to enhance image classification performance in diverse domains (Addepalli et al., 2024).

Recent efforts to compress and specialize multimodal models have led to techniques that reduce model size while maintaining strong performance on multimodal tasks (Fang et al., 2021). Some approaches enable multimodal generation by distilling vision-language knowledge (Dai et al., 2022), while others refine student models for specific applications through targeted distillation techniques (Wang et al., 2022). Additionally, a method incorporating affinity mimicking and weight inheritance has been proposed to compress CLIP models while preserving their strong zero-shot performance (Wu et al., 2023a).

Very recently, a Mixture-of-Visual-Encoder Knowledge Distillation (MoVE-KD) (Cao et al., 2025) was proposed to distill the unique proficiencies of multiple vision encoders into one efficient encoder model. A model Align-KD was proposed to guide the student model in VLM distillation to learn the cross-modal matching in the shallow layers (Feng et al., 2025). Several loss functions have been explored for CLIP distillation. Yang et al. (Yang et al., 2024) utilized ICL and MSE loss, while Li et al. (Li et al., 2024b) applied KL divergence for VLM distillation. In this work, we propose leveraging TE as a reward function to enhance VLM distillation.

## 3 INTRODUCTION TO TRANSFER ENTROPY

Transfer Entropy is an information-theoretic measure introduced by Schreiber (Schreiber, 2000) to quantify the directed transfer of information between two stochastic processes. It is particularly useful for detecting asymmetrical interactions and causal relationships, as it measures the influence that the past of one process, $X$, has on the future of another process, $Y$, beyond what can be explained by the past of $Y$ alone.

For two discrete-time stochastic processes $X(t)$ and $Y(t)$, the transfer entropy from $X$ to $Y$ is formally defined as (Schreiber, 2000):

$$T_{X \to Y} = \sum_t p(y_{t+1}, y_t, x_t) \log \frac{p(y_{t+1} \mid y_t, x_t)}{p(y_{t+1} \mid y_t)}, \tag{1}$$

where $p(\cdot)$ represents probability distributions of the respective random variables.

Transfer entropy is closely related to conditional mutual information. It can be rewritten as the conditional mutual information between $Y_{t+1}$ and $X_t$, conditioned on $Y_t$ (Shahsavari Baboukani et al., 2020):

$$T_{X \to Y} = I(Y_{t+1}; X_t \mid Y_t), \tag{2}$$

where $I(Y_{t+1}; X_t \mid Y_t)$ is the mutual information between $Y_{t+1}$ and the history of $X$, conditioned on the history of $Y$. The proof is provided in Appendix A. This formulation reveals that transfer entropy measures the additional information that $X_t$ provides about the future state $Y_{t+1}$, over and above the information provided by $Y$'s own history $Y_t$. In Appendix B, we present an overview of prior work on mutual information and TE estimation.

## 4 METHOD

### 4.1 WHY IS TRANSFER ENTROPY BENEFICIAL FOR VLM DISTILLATION

In the context of VLM distillation, we aim to maximize the information flow from a teacher model to a student model across both text and image modalities. To formalize this process, we define the following components:

- $V_t^{(T)}$: The teacher's intermediate visual representation (i.e., image features) at optimization step $t$.

- $S_t^{(T)}$: The teacher's intermediate textual representation at optimization step $t$.

- $V_t^{(S)}$: The student's current visual representation at optimization step $t$.

- $S_t^{(S)}$: The student's current textual representation at optimization step $t$.

- $V_{t+1}^{(S)}$: The student's updated visual representation at optimization step $t+1$ after incorporating guidance from the teacher.

- $S_{t+1}^{(S)}$: The student's updated textual representation at optimization step $t+1$ after incorporating guidance from the teacher.

The key idea is to quantify how much additional information from the teacher's guidance comprising both the teacher's textual representation $S_t^{(T)}$ and the teacher's visual features $V_t^{(T)}$ contributes to predicting the student's next states $V_{t+1}^{(S)}$ and $S_{t+1}^{(S)}$, beyond what is already present in the student's current states $V_t^{(S)}$ and $S_t^{(S)}$. This can be expressed using transfer entropy as

$$T_{(S_t^{(T)}, V_t^{(T)}) \rightarrow (S_t^{(S)}, V_t^{(S)})} = I\left((V_{t+1}^{(S)}, S_{t+1}^{(S)}); (V_t^{(T)}, S_t^{(T)}) \mid (V_t^{(S)}, S_t^{(S)})\right), \quad (3)$$

where $I(\cdot; \cdot \mid \cdot)$ denotes the conditional mutual information. This formulation measures the extent to which the teacher's combined text and image signals provide new information that drives the refinement of the student's textual and visual representations.

A high value of $T_{(S_t^{(T)}, V_t^{(T)}) \rightarrow (S_t^{(S)}, V_t^{(S)})}$ indicates that the teacher's guidance significantly influences the student's update. In the early stages of distillation, when the student's representations $V_t^{(S)}$ and $S_t^{(S)}$ are still underdeveloped, the influence of $S_t^{(T)}$ and $V_t^{(T)}$ is expected to be strong, resulting in a high transfer entropy. By analyzing $T_{(S_t^{(T)}, V_t^{(T)}) \rightarrow (S_t^{(S)}, V_t^{(S)})}$, we gain valuable insights into the balance and effectiveness of the information flow between the text and image modalities during distillation. Such insights can inform improvements in both the teacher's conditioning mechanism and the student's learning strategy, ultimately leading to more faithful and robust VLM distillation.

One might argue that since data samples within a batch are typically shuffled and independent, the assumption of temporal dependence between states may not hold. However, in (3), we do not interpret the index $t$ as wall-clock time or as referring to temporally correlated samples (e.g., in videos). Instead, $t$ represents the learning step of the student model, while the teacher remains fixed throughout training:

- Forward pass (step $t$): The teacher generates modality-specific hidden representations $(S_t^{(T)}, V_t^{(T)})$ for a given mini-batch. These remain constant, as the teacher's weights are frozen.

- Backward + parameter update: The student parameters are updated, producing new hidden states $(S_{t+1}^{(S)}, V_{t+1}^{(S)})$ for the same mini-batch during the next forward pass.

Therefore, TE is computed across optimization steps for the same data examples, rather than across different samples within a shuffled batch. This approach is consistent with prior work in information-theoretic analyses of learning dynamics (Goldfeld et al., 2019)(Achille & Soatto, 2018).

## 4.2 Approximating TE Using Cosine Similarity

In the context of VLM distillation, computing exact transfer entropy poses significant challenges due to the inherently high dimensionality of image and text representations. For example, in CLIP ResNet-50, the image and text embedding dimension is 1024 (Radford et al., 2021). Transfer entropy, a measure of the directed information flow between two systems, requires estimating conditional mutual information between high-dimensional feature spaces of teacher and student models. However, the joint distribution of image features (e.g., pixel-level data or patch embeddings) and text tokens (e.g., contextualized word embeddings) leads to an exponential increase in computational

complexity. This issue, often referred to as the curse of dimensionality (Köppen, 2000), renders exact computation of transfer entropy intractable. In (Gowri et al., 2025), it shows the difficulty of estimating mutual information in high dimensions. As represented in (2), TE is a conditional mutual information of two stochastic processes, which is more challenging. We propose the following Theorem to serve as the theoretical basis for approximating TE.

**Theorem 1** (First-order TE–Jacobian relation). *Let $x \in \mathbb{R}^d$ be an input (image–caption pair), and let $f_T, f_S : \mathbb{R}^d \to \mathbb{R}^D$ denote the teacher and student encoders with Jacobians*

$$J_T(x) = \nabla_x f_T(x), \quad J_S(x) = \nabla_x f_S(x) \in \mathbb{R}^{D \times d}. \tag{4}$$

*Under a first-order linear–Gaussian approximation of the conditional mutual information, the one-step transfer entropy from the teacher to the student satisfies*

$$T_T^S(x) \propto \cos\big(\widetilde{J}_S(x), \widetilde{J}_T(x)\big), \tag{5}$$

*where the Frobenius-normalized Jacobians are*

$$\widetilde{J}_S(x) = \frac{J_S(x)}{\|J_S(x)\|_F}, \qquad \widetilde{J}_T(x) = \frac{J_T(x)}{\|J_T(x)\|_F}, \tag{6}$$

*and $\cos(A, B) = \langle A, B \rangle_F$ denotes the cosine similarity (Frobenius inner product) between matrices A and B.*

In Appendix C, we provide the proof for this theorem.

In practice, we approximate the Jacobians using finite differences (Nocedal & Wright, 1999)(Baydin et al., 2018):

$$J_S \delta x \rightsquigarrow f_S(x + \delta x) - f_S(x), \qquad J_T \delta x \rightsquigarrow f_T(x + \delta x) - f_T(x),$$

where $\delta x$ is a small input perturbation. Based on the these theoretical results, we propose two approximations on TE using cosine similarity.

### 4.2.1 TE APPROXIMATION VIA COSINE SIMILARITY OF DIFFERENCES

Let $\mathbf{v}^{(S)}$ and $\mathbf{s}^{(S)}$ denote the image and text embeddings from the student model, and $\mathbf{v}^{(T)}$ and $\mathbf{s}^{(T)}$ denote the corresponding embeddings from the teacher model. The TE approximations are based on computing the cosine similarity between the directional changes in embeddings of the student and teacher models. The surrogate methods effectively captures how well the student follows the teacher's representation evolution.

To approximate TE, the method first calculates the difference between consecutive embeddings for both image and text modalities. This process assumes that batch ordering approximates temporal ordering, meaning that consecutive samples in the batch correspond to incremental states of representation learning. The embedding differences are computed as

$$\Delta\mathbf{v}_i^{(S)} = \mathbf{v}_{i+1}^{(S)} - \mathbf{v}_i^{(S)}, \quad \Delta\mathbf{v}_i^{(T)} = \mathbf{v}_{i+1}^{(T)} - \mathbf{v}_i^{(T)} \tag{7}$$

for images, and

$$\Delta\mathbf{s}_i^{(S)} = \mathbf{s}_{i+1}^{(S)} - \mathbf{s}_i^{(S)}, \quad \Delta\mathbf{s}_i^{(T)} = \mathbf{s}_{i+1}^{(T)} - \mathbf{s}_i^{(T)} \tag{8}$$

for text embeddings.

Once the differences are obtained, the next step involves computing the cosine similarity between the student's and teacher's directional changes. Cosine similarity (Xia et al., 2015) serves as a measure of alignment between the two models, ensuring that if the student's representation updates closely follow the teacher's, meaningful information transfer is occurring. The cosine similarity for images is given by

$$\cos\theta_i^{(v)} = \frac{\langle\Delta\mathbf{v}_i^{(S)}, \Delta\mathbf{v}_i^{(T)}\rangle}{\|\Delta\mathbf{v}_i^{(S)}\|\|\Delta\mathbf{v}_i^{(T)}\| + \epsilon}, \tag{9}$$

where $\epsilon$ is a small constant to prevent division by zero. While for text embeddings, it is given by

$$\cos\theta_i^{(s)} = \frac{\langle\Delta\mathbf{s}_i^{(S)}, \Delta\mathbf{s}_i^{(T)}\rangle}{\|\Delta\mathbf{s}_i^{(S)}\|\|\Delta\mathbf{s}_i^{(T)}\| + \epsilon}. \tag{10}$$

To approximate the overall TE for each modality, the method computes the mean cosine similarity across all batch elements. The image-based TE is computed as

$$TE_{\text{img}} = \frac{1}{B-1} \sum_{i=1}^{B-1} \cos \theta_i^{(v)}, \tag{11}$$

while the text-based TE follows the same formulation:

$$TE_{\text{txt}} = \frac{1}{B-1} \sum_{i=1}^{B-1} \cos \theta_i^{(s)}. \tag{12}$$

The final TE approximation is obtained by averaging the image and text TE values:

$$TE = \frac{1}{2} \left( TE_{\text{img}} + TE_{\text{txt}} \right). \tag{13}$$

### 4.2.2  TE Approximation via Cosine Similarity of Concatenated Differences

An alternative approach to approximate TE involves combining the directional changes from both image and text modalities before computing the cosine similarity. In this method, we first calculate the differences between consecutive embeddings for both modalities, same as (7)(8). Instead of computing the cosine similarity for each modality independently and then averaging the results, we concatenate the difference vectors from both modalities into a single vector. That is, for each index $i$, we define the concatenated difference vectors as

$$\Delta \mathbf{c}_i^{(S)} = \left[ \Delta \mathbf{v}_i^{(S)} \| \Delta \mathbf{s}_i^{(S)} \right], \tag{14}$$

$$\Delta \mathbf{c}_i^{(T)} = \left[ \Delta \mathbf{v}_i^{(T)} \| \Delta \mathbf{s}_i^{(T)} \right], \tag{15}$$

where $\|$ denotes concatenation along the feature dimension.

The cosine similarity between the concatenated difference vectors is then computed as

$$\cos \theta_i^{(\text{cat})} = \frac{\langle \Delta \mathbf{c}_i^{(S)}, \Delta \mathbf{c}_i^{(T)} \rangle}{\| \Delta \mathbf{c}_i^{(S)} \| \, \| \Delta \mathbf{c}_i^{(T)} \| + \epsilon}, \tag{16}$$

with $\epsilon$ being a small constant for numerical stability. Finally, the overall TE approximation is obtained by averaging these cosine similarities over all consecutive pairs in the batch:

$$TE = \frac{1}{B-1} \sum_{i=1}^{B-1} \cos \theta_i^{(\text{cat})}. \tag{17}$$

This concatenation-based surrogate for TE captures the joint evolution of image and text representations, providing a single metric that reflects how well the student model's combined modality updates align with those of the teacher.

In Appendix D, we present evaluation results for our two TE approximation methods and compare them with exact TE computation in simple experimental settings. In Appendix E, we analyze the computational cost of exact TE versus TE approximations.

### 4.3  Loss Functions for VLM Distillation

To effectively transfer knowledge from the teacher model to the student model in a VLM distillation setting, we employ a combination of Contrastive Loss (CL), KL divergence, MSE, ICL, and TE. The first four loss functions are introduced in Appendix F. These components ensure that the student model aligns its representations with the teacher while maintaining structural consistency across modalities.

To integrate the transfer entropy component, we subtract the surrogate TE reward from the overall loss. Combining these terms, the total loss function becomes:

$$\mathcal{L}_{\text{total}} = \mathcal{L}_{\text{contrastive}} + \alpha \, \mathcal{L}_{\text{KL}} + \beta \, \mathcal{L}_{\text{MSE}} + \delta \mathcal{L}_{\text{ICL}} - \gamma \, TE, \tag{18}$$

where $\alpha$, $\beta$, $\delta$ and $\gamma$ are weighting factors that balance the contributions of the KL divergence loss, the MSE loss, the ICL loss, and the TE reward, respectively. This composite loss encourages the student model to not only align with the teacher's predictions but also to capture the directional evolution of feature representations, resulting in more faithful distillation of multi-modal interactions.

By integrating CL, KL divergence, MSE loss, ICL loss, and TE-based regularization, we construct a comprehensive loss function that balances distributional alignment and information transfer, leading to a more effective VLM distillation process.

## 5 Experiments

Our experiments consist of the following configurations: (1) Teacher: ResNet-50, Student: ResNet-34; (2) Teacher: ViT-B/16, Student: ResNet-34; (3) Teacher: ResNet-50, Student: ResNet-18. We evaluate these settings on two datasets: MSCOCO 2014 (Lin et al., 2014) and Flickr8k (Hodosh et al., 2013)(Marco et al., 2023). We also include one application in classification based on Food 101 dataset (Bossard et al., 2014) in Appendix G.5.

In this Section, the teacher model employed in our experiments is OpenAI's CLIP RN50 (Radford et al., 2021), which integrates both an image encoder and a text encoder. The image encoder is based on a modified ResNet-50 architecture, comprising approximately 38.3 million parameters. The text encoder is a 12-layer Transformer (Vaswani et al., 2017) with a hidden dimension of 512, contributing around 63.1 million parameters (Radford et al., 2021). Combined, the CLIP RN50 model encompasses approximately 102 million parameters, positioning it as a moderately large-scale vision-language model well-suited for knowledge distillation tasks.

In contrast, the student model is based on RN34 for the image encoder and a lightweight Transformer for the text encoder. The RN34 architecture contains approximately 21.8 million parameters, and the final fully connected layer is modified to output 1024-dimensional features, keeping the parameter count relatively stable (He et al., 2016). The text encoder consists of an embedding layer with a vocabulary size of 49,408 and a hidden dimension of 1024, contributing approximately 25.3 million parameters (Mehta et al., 2020). Additionally, the student Transformer has only 2 encoder layers with an 8-head attention mechanism, leading to an estimated total of 5-10 million parameters (Vaswani et al., 2017). Combining both encoders, the total parameter count of the student model is approximately 55-60 million, significantly smaller than the teacher model while maintaining effective knowledge representation capabilities.

Our experiments are conducted on the MSCOCO 2014 dataset (Lin et al., 2014), which comprises approximately 82,783 training images and 40,504 validation images, each paired with multiple textual descriptions. This dataset is widely adopted in vision-language research due to its extensive and diverse image-caption pairs.

For clarity, we refer to the TE approximation introduced in Section 4.2.1 as TE1 and the approximation in Section 4.2.2 as TE2. The student model is trained using various combinations of loss functions, including Contrastive Loss (CL), KL divergence, MSE loss, ICL loss, and our proposed TE rewards. The total loss function is defined in (18). We conducted experiments using different combinations of these loss components. Our TE-based regularization is designed to capture the directional information flow between the teacher and student feature encoders, thereby encouraging the student to mimic the teacher's behavior more closely.

The hyperparameters $\alpha$, $\beta$, $\delta$, and $\gamma$ in (18) are designed to balance the contributions of the CL, KL, MSE, ICL, and TE terms in the overall objective. We assign larger values to hyperparameters associated with loss components that naturally exhibit smaller magnitudes, ensuring that each term contributes comparably to the optimization process. The training losses and TE for different loss functions are provided in Fig. 3.

In this experiment, we used weighting factors $\alpha = 1.0$, $\beta = 50$, $\delta = 1.0$, $\gamma = 1.0$, and a temperature parameter $\tau = 0.07$. These hyperparameters were carefully selected to balance the contributions of each loss component, ensuring effective knowledge transfer from the teacher to the student model while maintaining training stability. The batch size was set to $|B| = 64$, and the training data was shuffled to eliminate correlations between neighboring samples.

We utilized Google Colab Pro with a T4 GPU and High-RAM for training and performance evaluation. Due to time and budget constraints, we trained the student model, RN34, for only 10 epochs in each loss function combination scenario. The training and evaluation process for each experimental setup took approximately 14 hours.

Figure 3 illustrates that the total training loss decreases steadily over epochs while the TE rewards show an increasing trend. This behavior indicates that the model effectively minimizes the overall objective and progressively captures the directional information flow between teacher and student representations. The TE-based regularization plays a key role in maintaining structured alignment during training, which is critical for effective knowledge transfer. For experiments with TE1 and TE2 such as Fig. 3f and Fig. 3l, the TE1 and TE2 monotonically increase with very close but different values. However, the KL loss and MSE don't decrease clearly with more training epochs. In loss functions with KL, ICL, MSE, TE, different combinations may impact each other. For example, in Contrastive + TE1 (Fig. 3d), TE1 achieved average value 0.7242 at epoch 10; in Contrastive + KL + TE1 (Fig. 3g), TE1 achieved average value 0.7865 at epoch 10; and in Contrastive + KL + ICL + TE1 (Fig. 3j), TE1 achieved average value 0.4611 at epoch 10. So this demonstrated that KL promotes TE, but ICL discourages TE.

We evaluated the performance of the trained student models using Recall@k for both image-to-text (I2T) and text-to-image (T2I) retrieval tasks. Recall@k measures the percentage of queries for which the correct match appears in the top-k retrieved results (Manning et al., 2008). A higher Recall@1 indicates stronger alignment between images and texts, as the correct match is ranked first, while Recall@5 and Recall@10 provide insight into broader retrieval accuracy. We summarized the performance of the evaluation of the trained student model RN34 in Table 1.

Table 1: Comparison of zero-shot retrieval performance (Recall@k) of student RN34 with teacher RN50 for different loss function combinations in VLM distillation using **MSCOCO**. All Loss Function: CL + KL + MSE + ICL - TE1 - TE2.

| Model and Loss Function | I2T Retrieval (R) | | | T2I Retrieval (R) | | |
|---|---|---|---|---|---|---|
| | R@1 | R@5 | R@10 | R@1 | R@5 | R@10 |
| **Teacher Model (RN50)** | 15.27% | 30.73% | 39.05% | 11.68% | 25.52% | 33.50% |
| **Student Models (RN34)** | | | | | | |
| CL Only (Oord et al., 2018) | 4.94% | 14.60% | 22.51% | 3.96% | 12.67% | 19.45% |
| CL + MSE (Yang et al., 2024) | 5.13% | 15.41% | 23.17% | 4.00% | 12.79% | 19.53% |
| CL + KL (Li et al., 2024b) | 5.42% | 16.20% | 24.55% | 5.06% | 15.35% | 22.92% |
| CL + ICL (Yang et al., 2024) | 5.75% | 16.86% | 24.83% | 5.07% | 15.14% | 22.44% |
| CL - TE1 | 6.91% | 19.48% | 28.12% | 5.68% | 16.33% | 23.93% |
| CL - TE2 | 7.04% | 19.22% | 27.97% | 5.46% | 15.90% | 23.49% |
| CL - TE1 - TE2 | **8.24**% | **22.43**% | **31.73**% | 6.53% | 18.13% | 26.02% |
| CL + KL - TE1 | 7.81% | 21.31% | 30.65% | 6.42% | 18.18% | 26.33% |
| CL + KL - TE2 | 7.77% | 21.10% | 30.22% | 6.21% | 17.95% | 26.03% |
| CL + KL + MSE - TE1 | 7.62% | 21.05% | 30.34% | 6.53% | 18.48% | 26.66% |
| CL + KL + ICL - TE1 | 7.59% | 20.87% | 29.95% | 6.78% | 19.11% | 27.33% |
| CL + KL + MSE + ICL - TE1 | 7.51% | 20.62% | 29.81% | 6.76% | 19.02% | 27.32% |
| All Loss Function | 8.11% | 22.05% | 31.57% | **7.18**% | **19.75**% | **28.14**% |

Table 1 clearly demonstrates that incorporating transfer entropy (TE1 and TE2) into the VLM distillation objective leads to substantial performance gains in both I2T and T2I retrieval. We incrementally added each loss component to the base contrastive loss and observed that the introduction of TE1 or TE2 resulted in the most significant performance improvements. The best-performing models all include TE components, underscoring their effectiveness in enhancing the student model's ability to capture structured information flow from the teacher. Notably, the configuration using the loss function `CL - TE1 - TE2` achieves the highest I2T Recall@1, while the full loss combination `CL + KL + MSE + ICL - TE1 - TE2` yields the best T2I performance. This suggests that TE terms not only provide strong standalone regularization but also complement traditional distillation objectives when integrated holistically. As shown in Figure 3f, both TE1 and TE2 exhibit similar trends with monotonically increasing values during training. This indicates that their influ-

ence becomes more prominent over time, effectively guiding the optimization of the CL–TE1–TE2 objective. We highlight the best-performing scores in bold in Table 1.

Table 2 reports the sensitivity of retrieval performance to different hyperparameter settings. A clear trend emerges: the parameter $\gamma$, which controls the strength of the TE term, has a pronounced influence on the results. When $\gamma = 0$ (i.e., TE is omitted), performance drops sharply across both I2T and T2I tasks, highlighting the necessity of including TE in the loss. Introducing TE ($\gamma > 0$) consistently improves Recall@k, with different values favoring different tasks: smaller $\gamma$ yields the strongest T2I results, while larger $\gamma$ (e.g., 7.5) achieves the best I2T scores. This demonstrates that TE regularization is not only beneficial overall but also tunable for task-specific gains. In particular, moderate values of $\gamma$ strike a favorable balance, confirming that TE plays a critical role in enhancing the transfer of knowledge during distillation.

Table 2: Comparison of zero-shot retrieval performance (Recall@k) in percentage of student RN34 with teacher RN50 on MSCOCO.

| $\alpha$ | $\beta$ | $\delta$ | $\gamma$ | I2T R@1 | I2T R@5 | I2T R@10 | T2I R@1 | T2I R@5 | T2I R@10 |
|---|---|---|---|---|---|---|---|---|---|
| 1 | 50 | 1 | 0 | 6.08% | 18.14% | 26.93% | 5.92% | 17.06% | 24.89% |
| 1 | 50 | 1 | 1 | 8.11% | 22.05% | 31.57% | 7.18% | 19.75% | 28.14% |
| 1 | 50 | 1 | 2.5 | 9.48% | 24.68% | 34.54% | **7.57%** | **20.55%** | **29.03%** |
| 1 | 50 | 1 | 5 | 10.02% | 25.80% | 35.74% | 7.32% | 20.04% | 28.50% |
| 1 | 50 | 1 | 7.5 | **10.27%** | **26.36%** | **36.30%** | 6.98% | 18.95% | 26.95% |
| 1 | 50 | 1 | 10 | 9.77% | 25.25% | 35.04% | 6.94% | 18.78% | 26.67% |
| 5 | 50 | 1 | 5 | 10.19% | 26.15% | 36.09% | 6.81% | 18.84% | 26.88% |
| 1 | 100 | 1 | 5 | 10.25% | 25.76% | 35.69% | 7.30% | 20.55% | 29.03% |
| 1 | 50 | 5 | 5 | 8.20% | 22.14% | 31.63% | 7.14% | 19.61% | 28.05% |

In Appendix G, we present the experimental results for four more experiments: 1) teacher: RN50, student: RN34, dataset: Flick8k; 2) teacher: ViT-B/16, student: RN34; Datasets: MSCOCO and Flick8k; 3) teacher: RN50, student: RN18, dataset: MSCOCO; 4) teacher: RN50, student: RN34, dataset: Food-101, for classification. We evaluate zero-shot retrieval performance across multiple VLM distillation settings and find that incorporating transfer entropy (TE1 and TE2) consistently improves both I2T and T2I retrieval.

# 6 CONCLUSIONS AND FUTURE WORK

In this work, we introduced TE as a regularization technique for VLM distillation, aiming to enhance knowledge transfer from a teacher model to a student model. Direct computation of TE is intractable due to the high dimensionality of image and text representations. To address this, we demonstrated that a first-order (linear) expansion of TE yields a practical surrogate based on the cosine similarity between the Jacobians of the teacher and student processes. Building on this insight, we proposed two TE approximation strategies that leverage cosine similarity to quantify and enforce directional information flow between teacher and student embeddings across both image and text modalities. By integrating TE-based regularization into the distillation loss, we showed that the student model more effectively captures structured multimodal information, resulting in improved retrieval performance.

Our experiments were conducted using CLIP RN50 and ViT-B/16 as teacher models, and RN34 and RN18 as student models, evaluated on the MSCOCO 2014 and Flickr8k datasets. The experimental results underscore the importance of TE-based regularization for achieving improved feature alignment. Student models trained with TE consistently outperform those trained without TE, exhibiting notable gains in Recall@k for both image-to-text and text-to-image retrieval tasks.

This work primarily focuses on static teacher-student distillation, where the teacher model remains fixed during training. Future directions include extending our approach to co-distillation scenarios, wherein both teacher and student are jointly optimized to mutually enhance knowledge transfer. Additionally, exploring TE-based reinforcement learning strategies may provide an alternative optimization framework, enabling the student model to actively maximize meaningful information flow throughout training.

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

# A  RELATIONS BETWEEN TRANSFER ENTROPY, ENTROPY, AND MUTUAL INFORMATION

For two discrete-time stochastic processes $X(t)$ and $Y(t)$, the transfer entropy from $X$ to $Y$ is formally defined as (Schreiber, 2000):

$$T_{X \to Y} = \sum_t p(y_{t+1}, y_t, x_t) \log \frac{p(y_{t+1} \mid y_t, x_t)}{p(y_{t+1} \mid y_t)}, \tag{19}$$

where $p(\cdot)$ represents probability distributions of the respective random variables.

Transfer entropy can also be expressed in terms of conditional entropy and mutual information (Shahsavari Baboukani et al., 2020). Specifically, the transfer entropy from $X$ to $Y$, denoted $T_{X \to Y}$, measures the reduction in uncertainty about the future state $Y_{t+1}$ given the joint past of $X$ and $Y$, compared to the uncertainty given the past of $Y$ alone. Based on (19), this difference can be expressed as:

$$
\begin{aligned}
T_{X \to Y} &= \sum_t p(y_{t+1}, y_t, x_t) \log \frac{1}{p(y_{t+1} \mid y_t)} + \\
&\quad \sum_t p(y_{t+1}, y_t, x_t) \log p(y_{t+1} \mid y_t, x_t) \\
&= \sum_t p(y_{t+1}, y_t) \log \frac{1}{p(y_{t+1} \mid y_t)} + \\
&\quad \sum_t p(y_{t+1}, y_t, x_t) \log p(y_{t+1} \mid y_t, x_t) \\
&= H(Y_{t+1} \mid Y_t) - H(Y_{t+1} \mid Y_t, X_t) \tag{20} \\
&= I(Y_{t+1}; X_t \mid Y_t) \tag{21}
\end{aligned}
$$

where $H(Y_{t+1} \mid Y_t)$ is the conditional entropy of $Y_{t+1}$ given its own history $Y_t$, $H(Y_{t+1} \mid Y_t, X_t)$ is the conditional entropy of $Y_{t+1}$ given both the history of $Y$ and the history of $X$, and $I(Y_{t+1}; X_t \mid Y_t)$ is the mutual information between $Y_{t+1}$ and the history of $X$, conditioned on the history of $Y$. In this formulation, the transfer entropy quantifies the amount by which the uncertainty about the future of $Y$ is reduced by incorporating information from $X$.

# B  PRIOR WORK ON MUTUAL INFORMATION AND TRANSFER ENTROPY ESTIMATION

Mutual Information (MI) techniques have been employed to capture shared information between variables (Hjelm et al., 2018)(Oord et al., 2018). MINE (Belghazi et al., 2018) offers a differentiable estimator for mutual information, and information-theoretic regularization has been applied in generative models for disentanglement and improved control (Chen et al., 2016). In (Gao et al., 2015), a mutual information estimator was proposed based on modified k-nearest neighbor (KNN) that is robust to local non-uniformity with limited data. A diverse set of distributions with known MI values were introduced to evaluate the performance of different MI estimators beyond traditional normal distributions (Czyż et al., 2023). McAllester and Stratos (McAllester & Stratos, 2020) highlighted the inherent difficulties in estimating mutual information from finite data, demonstrating that any distribution-free high-confidence lower bound on MI cannot exceed $O(\ln N)$, thereby underscoring the fundamental challenges in accurate mutual information estimation without strong assumptions about the data distribution. Goldfeld and Greenewald (Goldfeld & Greenewald, 2021) introduced Sliced Mutual Information, a scalable measure that projects high-dimensional distributions onto one-dimensional subspaces, effectively capturing complex dependencies while reducing computational complexity. Approximating mutual information of high-dimensional variables using learned representations was studied in (Gowri et al., 2025).

Transfer entropy is a conditional mutual information from two stochastic processes, so it's more challenging in TE estimation. In (Zhang, 2018), Low-dimensional approximation in the searching procedure was applied to transfer entropy from non-uniform embedding. In (Zhu et al., 2015), KNN

was used for TE estimation. However, KNN-based approach doesn't work well if the data are noisy and long ranged. To overcome this weakness, a perturbation model based on locality sensitive hash function was proposed for TE estimation (Garg et al., 2022). Three estimators were used for TE estimation (Lee et al., 2012), namely fixed-binning with ranking, kernel density estimation, and the Darbellay-Vajda (D-V) adaptive partitioning algorithm extended to three dimensions. In (Ma, 2019), copula entropy was applied to TE estimation. To overcome the curse of dimensionality in TE estimation, TE was decomposed into a sum of finite-dimensional contributions in (Runge et al., 2012). Recently, transformer was used for TE estimation (Luxembourg et al., 2024). In this paper, we propose TE approximation approaches which can tremendously reduce the computation cost and overcome the curse of dimensionality.

## C  PROOF OF THEOREM 1

This section shows how a first-order (linear) expansion of TE leads to a computable surrogate based on a cosine similarity between the teacher– and student-process Jacobians. Our derivation follows the *linear–Gaussian surrogate* technique proposed in (Goldfeld et al., 2019).

*Proof.* Let $x \in \mathbb{R}^d$ be an input image–caption pair, and $f_T(x)$, $f_S(x) \in \mathbb{R}^D$ denote the teacher and student embeddings, respectively. Denote their Jacobians as $J_T(x) = \nabla_x f_T(x)$ and $J_S(x) = \nabla_x f_S(x)$, both in $\mathbb{R}^{D \times d}$.

To study the local behavior around $x$, consider a small perturbation $\delta x \sim \mathcal{N}(0, \sigma^2 I_d)$, and define

$$u := f_T(x + \delta x), \qquad v_t := f_S(x), \qquad v_{t+1} := f_S(x + \delta x).$$

The one-step transfer entropy from teacher to student becomes:

$$T_T^S(x) = I\big(v_{t+1};\, u \,|\, v_t\big). \tag{22}$$

Using a first-order Taylor expansion around $x$:

$$u \approx u_0 + J_T \delta x, \quad v_{t+1} \approx v_0 + J_S \delta x, \quad v_t = v_0 = f_S(x), \tag{23}$$

where $u_0 = f_T(x)$. Since $v_0$ is a constant shift, subtracting it from both sides does not change the conditional mutual information. Therefore:

$$T_T^S(x) \approx I\big(J_S \delta x;\, J_T \delta x\big). \tag{24}$$

Because $\delta x \sim \mathcal{N}(0, \sigma^2 I_d)$ and both $J_S$ and $J_T$ are linear maps, the pair $(J_S \delta x,\, J_T \delta x)$ is jointly Gaussian. Define the covariances:

$$\Sigma_S = \sigma^2 J_S J_S^\top, \quad \Sigma_T = \sigma^2 J_T J_T^\top, \quad \Sigma_{ST} = \sigma^2 J_S J_T^\top.$$

The mutual information between jointly Gaussian vectors is (Cover, 1999):

$$
\begin{aligned}
I\big(J_S \delta x;\, J_T \delta x\big) &= h\big(J_S \delta x\big) + h\big(J_T \delta x\big) - h\big(J_S \delta x,\, J_T \delta x\big) & (25) \\
&= \frac{1}{2} \log \frac{\det \Sigma_S \ \det \Sigma_T}{\det \begin{pmatrix} \Sigma_S & \Sigma_{ST} \\ \Sigma_{TS} & \Sigma_T \end{pmatrix}} & (26) \\
&= -\frac{1}{2} \log \det \left( I - \Sigma_S^{-1/2} \Sigma_{ST} \Sigma_T^{-1/2} \right). & (27)
\end{aligned}
$$

If $\Sigma_{ST}$ is small compared to the product $\Sigma_S^{1/2} \Sigma_T^{1/2}$ (which is often true in early training), we can use the approximation $\log \det(I - A) \approx -\operatorname{tr}(A)$ (Magnus & Neudecker, 1999). This gives (Goldfeld et al., 2019):

$$T_T^S(x) \approx \frac{\sigma^2}{2} \operatorname{tr}\left( (J_S J_S^\top)^{-1/2} J_S J_T^\top (J_T J_T^\top)^{-1/2} \right). \tag{28}$$

We can normalize both Jacobians by their Frobenius norms:

$$\widetilde{J}_S = \frac{J_S}{\|J_S\|_F}, \qquad \widetilde{J}_T = \frac{J_T}{\|J_T\|_F},$$

so that equation equation 28 becomes:

$$T_T^S(x) \propto \langle \widetilde{J}_S, \ \widetilde{J}_T \rangle_F = \cos\left( \widetilde{J}_S, \ \widetilde{J}_T \right), \tag{29}$$

i.e., the Frobenius inner product (cosine similarity) of the two Jacobians.

$\square$

# D   PERFORMANCE COMPARISON: TE APPROXIMATIONS VERSUS EXACT TE

We evaluated our two approximations of TE in Section 4.2 against the exact TE computed from a synthetic Gaussian channel. Specifically, teacher embeddings $\mathbf{T} \in \mathbb{R}^D$ are sampled from a standard normal distribution $\mathbf{T} \sim \mathcal{N}(0, I)$, and student embeddings are generated as

$$\mathbf{S} = \alpha \, \mathbf{T} + \sqrt{1 - \alpha^2} \, \mathbf{N}, \tag{30}$$

where $\mathbf{N} \sim \mathcal{N}(0, I)$ and $\alpha \in [0, 0.99]$ controls the teacher–student correlation. So each corresponding pair of teacher and student components forms a jointly Gaussian random pair with Pearson correlation coefficient $\alpha$ (Lee Rodgers & Nicewander, 1988). It is a classical result in information theory that for two jointly Gaussian random variables $X$ and $Y$ with correlation $\alpha$, the mutual information is given by (Cover, 1999)

$$I(X; Y) = -\frac{1}{2} \log(1 - \alpha^2). \tag{31}$$

In our setting, the exact transfer entropy is defined as (Shahsavari Baboukani et al., 2020)

$$\mathrm{TE}_{\mathrm{exact}} = I(Y_{t+1}; X_t \mid Y_t), \tag{32}$$

where $Y_{t+1}$ represents the student's updated representation, $X_t$ is the teacher's representation at time $t$, and $Y_t$ is the student's current representation. Under the common assumption that these variables are jointly Gaussian and the update of $Y_{t+1}$ depends linearly on $X_t$ (after conditioning on $Y_t$), a closed-form expression for the conditional mutual information can be derived. In particular, if the effective correlation between $X_t$ and $Y_{t+1}$ (after accounting for $Y_t$) is given by $\alpha$, then the mutual information per embedding dimension becomes

$$I(Y_{t+1}; X_t \mid Y_t) = -\frac{1}{2} \log(1 - \alpha^2). \tag{33}$$

When the embeddings have $D$ independent dimensions, this yields

$$\mathrm{TE}_{\mathrm{exact}} = \frac{D}{2} \, \log\left( \frac{1}{1 - \alpha^2} \right). \tag{34}$$

For ease of comparison with our cosine similarity–based approximations, we further normalize this exact TE value via a logarithmic transformation to map it into the interval $[0, 1]$ using the following transformation (Han et al., 2012):

$$\mathrm{TE}_{\mathrm{norm}} = \frac{\log(1 + \mathrm{TE}_{\mathrm{exact}})}{\log(1 + \mathrm{TE}_{\mathrm{max}})}, \tag{35}$$

where $\mathrm{TE}_{\mathrm{max}}$ is computed using $\alpha_{\mathrm{max}} = 0.99$ to define the upper bound for normalization.

For the two approximation methods proposed in Sections 4.2.1 (Method 1) and 4.2.2 (Method 2), we conducted experiments by varying $\alpha$ from 0 to 0.99, and computed the two approximation results and the normalized exact TE. The results are summarized in Fig. 1. The Pearson correlation between the normalized exact TE and both TE approximations was found to be 0.994, indicating a very strong linear relationship. These findings suggest that both approximation methods reliably track the exact TE, capturing the relative information flow from the teacher to the student in this synthetic setting.

We also examined the robustness of our two TE approximation methods as we varied two key factors in a synthetic teacher–student setting:

- Batch size ($B$), which affects the stability of sample-based estimates.

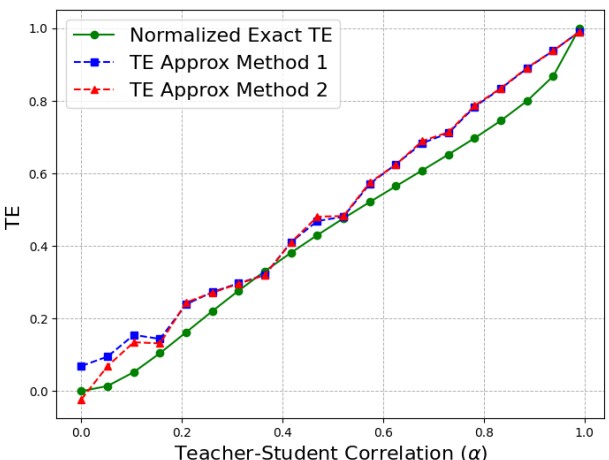

Figure 1: Comparison of TE Approximations vs. Normalized Exact TE.

• Embedding dimension ($D$), which influences the amount of representational capacity.

We fixed the teacher–student correlation coefficient at $\alpha = 0.8$ in (30). Two separate experiments were performed:

1. Varying batch size: We fix $D = 500$ and consider batch sizes $B \in \{10, 20, 50, 100, 200, 500, 1000\}$.

2. Varying embedding dimension: We fix $B = 500$ and let $D \in \{10, 20, 50, 100, 200, 500, 1000\}$.

In both cases, we computed the TE Approximation Method 1 and Method 2, and the normalized exact TE.

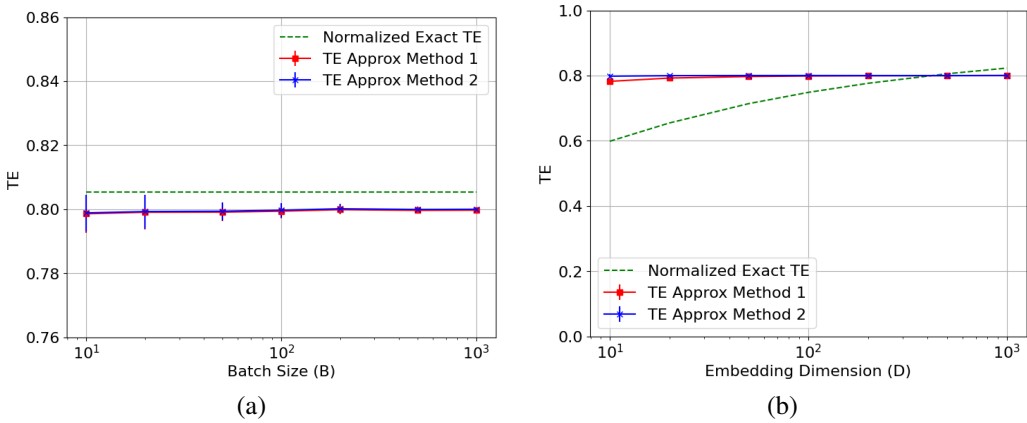

Figure 2: (a) TE approximations versus batch size ($B$) at fixed $D = 500$. (b) TE approximations versus embedding dimension ($D$) at fixed $B = 500$.

Figure 2a shows the behavior of these metrics as a function of batch size. Observe that both approximation methods rapidly converge to a stable estimate near the normalized exact TE (green dashed line). For very small $B$ (around 10–20), the sample-based cosine measures show slight deviations but still remain close to the exact TE. As $B$ grows, the variance diminishes and both approximations tightly match the theoretical reference.

Figure 2b illustrates the impact of varying embedding dimension $D$. Since the *exact* TE increases with $D$ (due to additional degrees of freedom), its normalized value (green line) also increases. By contrast, the two TE approximations remain relatively stable, hovering around 0.75–0.80 for all tested dimensions. This highlights a key property of the approximate measures: they capture the relative alignment between teacher and student (controlled by $\alpha$), but they do not grow with the embedding dimensionality as the exact mutual information does. In practice, this makes them computationally efficient and robust to high-dimensional data, though they are not designed to quantify the absolute amount of information transferred. Overall, these results confirm that both approximation methods track the ground-truth TE trend (in terms of relative comparisons), while offering a simpler and more scalable alternative to exact TE in high-dimensional settings.

The underlying intuition behind these approximation is that if the student's directional changes closely match those of the teacher, then information transfer is effectively occurring. Traditional transfer entropy measures rely on probability distributions over time, but this approach circumvents such computational overhead by leveraging geometric similarity in embedding space. By treating the batch as a sequence of evolving representations, the method estimates how well the teacher's influence propagates to the student. However, unlike traditional TE, which explicitly models information transfer through probability distributions, our approximation purely relies on directional alignment. Additionally, equal weighting of image and text modalities may not always be ideal, as one modality may contribute more to the learning process than the other.

The cosine-based TE approximations are highly effective in capturing the relative information flow in VLM distillation – they are easy to compute, robust across high dimensions, and correlate well with true information transfer. They confirm the intuition that a student embedding space matching the teacher's geometry is a good sign of successful knowledge distillation. However, these approximations do not measure exact information volume. They compress the notion of "how much knowledge" into a bounded similarity score. As a result, they are best used for comparing models or monitoring training (where the scale can be assumed fixed and only relative changes matter) rather than for absolute information quantification.

# E   COMPUTATIONAL COST ANALYSIS: EXACT TE VERSUS TE APPROXIMATIONS

**Computational Complexity:**

Exact TE often involves $O(N^2 d)$ operations, where $N$ is the number of samples and $d$ is the feature dimension. This is due to the need for joint probability estimation over multiple variables, which scales poorly as data size and dimensionality increase. In contrast, TE Approximation Method 1 in Section 4.2.1 uses cosine similarity to estimate TE by focusing on local neighborhoods (difference of neighbors) in the embedding space. Instead of constructing a full joint probability table, for each observation one can find a set of "similar" past states (e.g. nearest neighbors in terms of cosine distance) and approximate conditional probabilities from those neighbors. The neighbor-based cosine similarity approximation reduces complexity by considering only local neighborhoods in the embedding space. By focusing on a limited number of similar past states instead of the entire dataset, this method lowers the computational cost to approximately $O(N \log N)$ with efficient neighbor searches. TE Approximation Method 2 in 4.2.2 uses cosine similarity to concatenate the high-dimensional states, thereby reducing the state space before computing TE. The concatenation-based approximation further reduces complexity by grouping similar data points into clusters and treating each cluster as a discrete state, leading to an effective time complexity of $O(Nd)$ for the clustering process and $O(N)$ for TE calculation.

**Memory Usage:** Exact TE requires storing large joint probability distributions, which grow exponentially with dimensionality. This makes exact computation infeasible for high-dimensional embeddings, as it demands large storage space for probability tables or expensive nearest-neighbor searches. The approximations mitigate this issue by avoiding explicit density estimation. The TE Approximation Method 1 only stores similarity measures and a small set of neighbors for each data point, keeping memory usage at $O(Nd)$. The TE Approximation Method 2 concatenates data into a limited number of clusters, further reducing storage requirements to $O(Cd)$, where $C$ is the number of clusters, much smaller than $N$. These approximations thus enable TE computation in large-scale deep learning applications without overwhelming memory constraints.

**Scalability in High Dimensions:** Exact TE suffers from the curse of dimensionality. As dimensionality increases, joint probability estimation becomes unreliable because high-dimensional data points become sparse, making density estimation difficult. This often results in TE values that are biased towards zero. In contrast, cosine similarity-based methods are much more scalable, as cosine similarity is well-defined even in high dimensions and can be computed efficiently. The TE Approximate method 1 relies on approximate nearest-neighbor searches, which remain feasible even as $d$ grows, while the TE Approximate method 2 concatenates high-dimensional data into a manageable number of clusters, making TE estimation practical even for very large embeddings.

In summary, using cosine similarity approximations for transfer entropy enables analysis of high-dimensional and large-scale data that would be otherwise impossible with exact methods. The approaches significantly improve computational feasibility and can even enhance statistical reliability in data-limited situations (Zhang, 2018). The cost, however, is that we must accept an approximate measure that may overlook complex nuances of the data's information dynamics. Since the priority in CLIP is to handle very rich embeddings and get a fast, actionable estimate of information flow, the TE approximation methods are invaluable.

# F  LOSS FUNCTIONS IN VLM DISTILLATION

## F.1  LOGIT REPRESENTATION IN VLM DISTILLATION

In our framework, logits represent the similarity scores between image and text embeddings, which are fundamental to contrastive learning. Given a batch of image-text pairs, let $\mathbf{v}^{(S)}, \mathbf{s}^{(S)}$ denote the image and text embeddings from the student model, and $\mathbf{v}^{(T)}, \mathbf{s}^{(T)}$ denote the corresponding embeddings from the teacher model. The logit computation follows these steps.

First, we normalize the embeddings to unit norm:

$$\hat{\mathbf{v}}^{(S)} = \frac{\mathbf{v}^{(S)}}{\|\mathbf{v}^{(S)}\|_2}, \quad \hat{\mathbf{s}}^{(S)} = \frac{\mathbf{s}^{(S)}}{\|\mathbf{s}^{(S)}\|_2}, \tag{36}$$

$$\hat{\mathbf{v}}^{(T)} = \frac{\mathbf{v}^{(T)}}{\|\mathbf{v}^{(T)}\|_2}, \quad \hat{\mathbf{s}}^{(T)} = \frac{\mathbf{s}^{(T)}}{\|\mathbf{s}^{(T)}\|_2}. \tag{37}$$

The similarity logits for the student and teacher models are then computed as the dot product between the corresponding image and text embeddings, scaled by a temperature parameter $\tau$:

$$\mathbf{z}^{(S)} = \frac{\hat{\mathbf{v}}^{(S)} \cdot (\hat{\mathbf{s}}^{(S)})^\top}{\tau}, \quad \mathbf{z}^{(T)} = \frac{\hat{\mathbf{v}}^{(T)} \cdot (\hat{\mathbf{s}}^{(T)})^\top}{\tau}. \tag{38}$$

Here, $\mathbf{z}^{(S)}$ and $\mathbf{z}^{(T)}$ are $|B| \times |B|$ matrices, where each entry $z_{ij}^{(S)}$ represents the similarity between the $i$-th image embedding and the $j$-th text embedding in the batch for the student model, and similarly for the teacher model. The temperature parameter $\tau$ controls the sharpness of the similarity distribution, with lower values making the distribution more peaky.

These logits are subsequently used in the contrastive loss and KL divergence computation to align the student's feature representations with those of the teacher, ensuring effective knowledge transfer during distillation. Several studies have explored the computation and utilization of these logits in image-text contrastive frameworks (Radford et al., 2021)(Jia et al., 2021)(Yang et al., 2022)(Hasegawa et al., 2023)(Xiao et al., 2024).

## F.2  CONTRASTIVE LOSS FOR VLM DISTILLATION

We employ a contrastive loss based on the InfoNCE loss formulation to align the student model's image and text representations effectively. Given a batch of $|B|$ image-text pairs, we define the contrastive loss using the computed logits. The contrastive loss for image-to-text alignment is defined as (Oord et al., 2018) (Yang et al., 2024):

$$\mathcal{L}_{I \to T} = -\frac{1}{|B|} \sum_{k=1}^{|B|} \log \frac{\exp(z_{kk}^{(S)})}{\sum_{j=1}^{|B|} \exp(z_{kj}^{(S)})} \tag{39}$$

$z_{kj}^{(S)}$ represents the similarity between the $k$-th image embedding and the $j$-th text embedding in the batch for the student model, and $z_{kk}^{(S)}$ represents the similarity logit between the $k$-th image and its corresponding text in the batch for the student model.

Similarly, the contrastive loss for text-to-image alignment is given by:

$$\mathcal{L}_{T \to I} = -\frac{1}{|B|} \sum_{k=1}^{|B|} \log \frac{\exp(z_{kk}^{(S)})}{\sum_{j=1}^{|B|} \exp(z_{jk}^{(S)})} \tag{40}$$

$z_{jk}^{(S)}$ represents the similarity between the $j$-th image embedding and the $k$-th text embedding in the batch for the student model, and $z_{kk}^{(S)}$ is the same as that in (39).

The total contrastive loss, which balances both image-to-text and text-to-image objectives, is computed as:

$$\mathcal{L}_{\text{contrastive}} = \frac{1}{2}(\mathcal{L}_{I \to T} + \mathcal{L}_{T \to I}). \tag{41}$$

This loss function encourages the student model to align its multi-modal representations by bringing matching pairs closer in the embedding space while pushing apart non-matching pairs. Contrastive loss has been extensively applied to knowledge distillation (Tian et al., 2019)(Chen et al., 2021)(Gao et al., 2021)(Peng et al., 2022)(Zhu et al., 2021)(Guo et al., 2023).

To enhance the effectiveness of distillation, we extend this contrastive loss with additional terms such as KL divergence and transfer entropy-based regularization. These terms further refine the student model's learning dynamics by ensuring information flow from the teacher's embeddings to the student's representations while preserving structural consistency across modalities.

### F.3 KL DIVERGENCE FOR VLM DISTILLATION

To ensure that the student model effectively mimics the probability distributions of the teacher model, we include a Kullback-Leibler (KL) divergence loss term. KL divergence measures how much the student's predicted distribution deviates from the teacher's distribution, enforcing a closer alignment between their logits. KL divergence has been applied to VLM distillation (Li et al., 2024b)(Sun et al., 2024).

For a given batch of image-text pairs, let $\mathbf{z}^{(S)}$ and $\mathbf{z}^{(T)}$ represent the similarity logits of the student and teacher models, respectively. The soft probability distributions are obtained via the softmax function:

$$P_i^{(S)} = \frac{\exp(z_i^{(S)}/\tau)}{\sum_{j=1}^{|B|} \exp(z_j^{(S)}/\tau)}, \tag{42}$$

$$P_i^{(T)} = \frac{\exp(z_i^{(T)}/\tau)}{\sum_{j=1}^{|B|} \exp(z_j^{(T)}/\tau)}, \tag{43}$$

where $\tau$ is the temperature parameter that controls the sharpness of the distributions.

The KL divergence loss is computed as:

$$\mathcal{L}_{\text{KL}} = \frac{1}{2} \left( D_{\text{KL}} \left( P_{\text{image}}^{(S)} \| P_{\text{image}}^{(T)} \right) + D_{\text{KL}} \left( P_{\text{text}}^{(S)} \| P_{\text{text}}^{(T)} \right) \right), \tag{44}$$

where the KL divergence between two probability distributions $P^{(S)}$ and $P^{(T)}$ is defined as:

$$D_{\text{KL}}(P^{(S)} \| P^{(T)}) = \sum_{i=1}^{|B|} P_i^{(T)} \log \frac{P_i^{(T)}}{P_i^{(S)}}. \tag{45}$$

This loss encourages the student model to produce probability distributions that closely resemble those of the teacher, effectively preserving the knowledge distilled from the teacher while allowing the student to generalize efficiently.

### F.4 MSE Loss Function for VLM Distillation

To further align the feature representations of the teacher and student models, we include MSE loss that minimizes the discrepancy between their intermediate embeddings (Yang et al., 2024). The MSE loss is computed as the sum of the squared differences between the student and teacher embeddings for both modalities:

$$\mathcal{L}_{\text{MSE}} = \mathcal{L}_{\text{MSE}}^{\text{image}} + \mathcal{L}_{\text{MSE}}^{\text{text}}, \tag{46}$$

where

$$\mathcal{L}_{\text{MSE}}^{\text{image}} = \frac{1}{|B|} \sum_{i=1}^{|B|} \left\| \hat{\mathbf{v}}_i^{(S)} - \hat{\mathbf{v}}_i^{(T)} \right\|^2, \tag{47}$$

$$\mathcal{L}_{\text{MSE}}^{\text{text}} = \frac{1}{|B|} \sum_{i=1}^{|B|} \left\| \hat{\mathbf{s}}_i^{(S)} - \hat{\mathbf{s}}_i^{(T)} \right\|^2. \tag{48}$$

Here, $|B|$ represents the batch size, and $\|\cdot\|^2$ denotes the squared Euclidean norm. This loss ensures that the student model's learned embeddings remain close to the teacher's representations in the feature space, facilitating effective knowledge transfer. MSE has been applied to VLM loss function in (Yang et al., 2024), and was called feature distillation.

### F.5 Interactive Contrastive Learning

Interactive Contrastive Learning (ICL) was proposed in (Yang et al., 2024) to aligns the student model's feature representations with those of the teacher by treating the student embeddings as anchors and contrasting them with the teacher embeddings.

Given a batch of image-text pairs, let $\mathbf{v}_k^{(S)}$ be the image embedding from the student model, and $\{\mathbf{s}_b^{(T)}\}_{b=1}^{|B|}$ denote the contrastive text embeddings from the teacher model. The image-to-text ICL loss is formulated as:

$$\mathcal{L}_{\text{ICL}}^{I \to T} = -\log \frac{\exp(\mathbf{v}_k^{(S)} \cdot \mathbf{s}_k^{(T)} / \tau)}{\sum_{b=1}^{|B|} \exp(\mathbf{v}_k^{(S)} \cdot \mathbf{s}_b^{(T)} / \tau)}, \tag{49}$$

where $\tau$ is the temperature parameter.

Similarly, for a student text embedding $\mathbf{s}_k^{(S)}$ and contrastive image embeddings from the teacher model $\{\mathbf{v}_b^{(T)}\}_{b=1}^{|B|}$, the text-to-image ICL loss is:

$$\mathcal{L}_{\text{ICL}}^{T \to I} = -\log \frac{\exp(\mathbf{s}_k^{(S)} \cdot \mathbf{v}_k^{(T)} / \tau)}{\sum_{b=1}^{|B|} \exp(\mathbf{s}_k^{(S)} \cdot \mathbf{v}_b^{(T)} / \tau)}. \tag{50}$$

The final ICL loss is a combination of the two:

$$\mathcal{L}_{\text{ICL}} = \frac{1}{2} \left( \mathcal{L}_{\text{ICL}}^{I \to T} + \mathcal{L}_{\text{ICL}}^{T \to I} \right). \tag{51}$$

By integrating ICL, the student model effectively learns from the teacher's structured feature space, leading to improved representation learning and knowledge transfer.

## G More Experimental Results

### G.1 Teacher: RN50, Student: RN34, Dataset: MSCOCO

Figure 3 shows the training losses and TE rewards over epochs for various configurations in the experiment with teacher RN50 and student RN34.

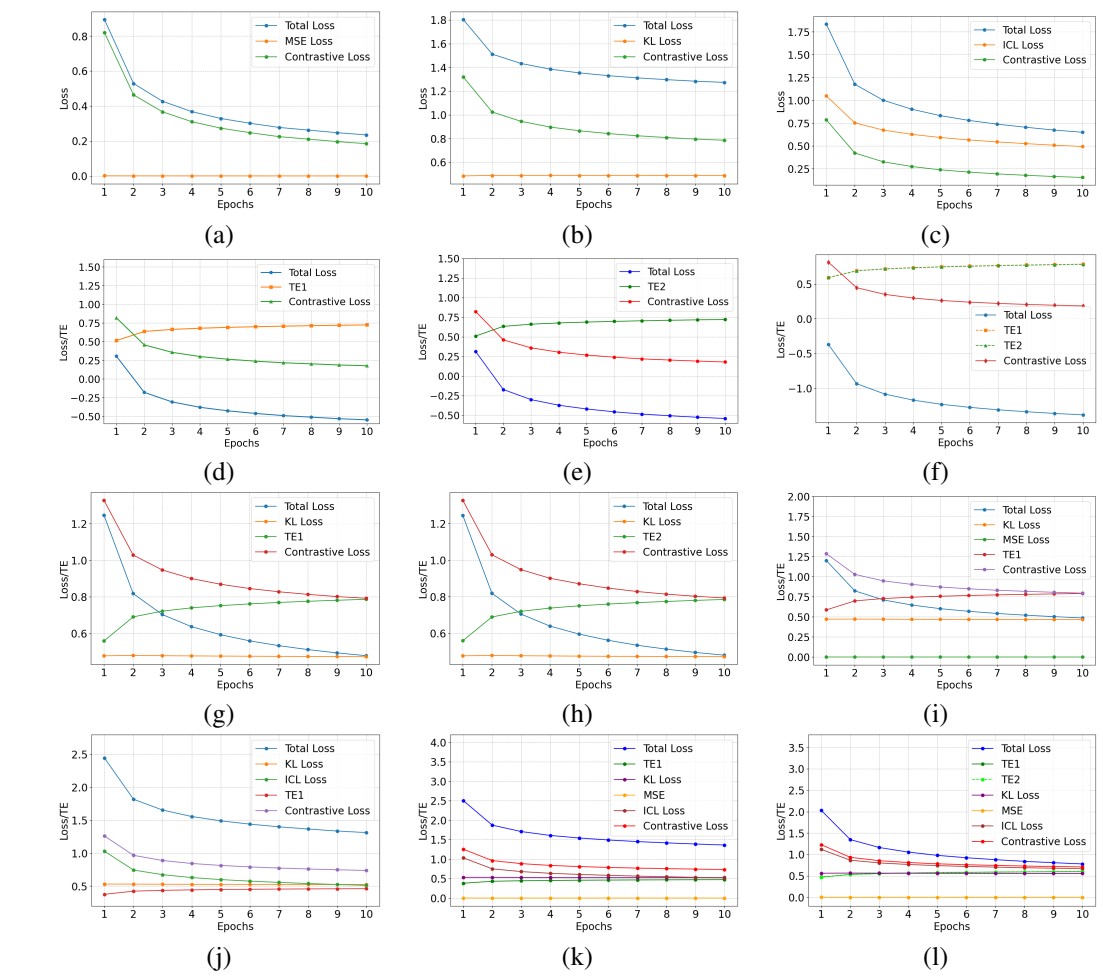

Figure 3: The training losses and TE for different loss functions in the training of Student Model RN34 using MSCOCO dataset. (a) Contrastive + MSE, (b) Contrastive + KL, (c) Contrastive + ICL, (d) Contrastive - TE1, (e) Contrastive - TE2, (f) Contrastive - TE1 - TE2, (g) Contrastive + KL - TE1, (h) Contrastive + KL - TE2, (i) Contrastive + KL + MSE - TE1, (j) Contrastive + KL + ICL - TE1, (k) Contrastive + KL + ICL +MSE - TE1, (l) Contrastive + KL + ICL +MSE -TE1 - TE2.

### G.2   TEACHER: RN50, STUDENT: RN34, DATASET: FLICK8K

We further evaluate our approach on the Flickr8k dataset (Marco et al., 2023), using 85% of the data for training and 15% for testing. Performance results for various loss functions are summarized in Table 3. The loss function employs weighting factors $\alpha = 1.0$, $\beta = 100$, $\delta = 1.0$, $\gamma = 5.0$, and a temperature parameter $\tau = 0.07$. These parameters were selected based on the relative contribution of each loss term to the total loss during training, ensuring balanced optimization. Given the modest size of Flickr8k, all experiments were conducted on a Google Colab instance equipped with an A100 GPU and limited system RAM. Each experiment (i.e., each row in Table 3) required approximately 20 minutes of training time. Notably, incorporating TE1 or TE2 into the loss function consistently improves both image-to-text (I2T) and text-to-image (T2I) retrieval performance compared to baselines that rely solely on standard distillation losses such as CL + KL or CL + MSE. These results underscore the effectiveness of transfer entropy approximations in guiding student model updates during distillation.

Table 4 presents the sensitivity analysis of the hyperparameters $\alpha$, $\beta$, $\delta$, and $\gamma$ in the loss function (18) for zero-shot retrieval on Flickr8k. A consistent trend emerges: setting $\gamma = 0$ (i.e., omitting the TE term) leads to notably lower performance across both I2T and T2I tasks. By contrast, introducing

Table 3: Zero-shot retrieval performance (Recall@k) on **Flickr8k** of student RN34 using teacher RN50 under different loss functions. All Loss Function: CL + KL + MSE + ICL - TE1 - TE2.

| Model and Loss Function | I2T Retrieval (R) | | | T2I Retrieval (R) | | |
|---|---|---|---|---|---|---|
| | R@1 | R@5 | R@10 | R@1 | R@5 | R@10 |
| **Teacher Model (RN50)** | 51.65% | 78.17% | 87.73% | 47.28% | 75.21% | 84.60% |
| **Student Models (RN34)** | | | | | | |
| CL Only (Oord et al., 2018) | 22.73% | 48.19% | 60.87% | 18.47% | 43.76% | 56.77% |
| CL + MSE (Yang et al., 2024) | 22.98% | 49.92% | 62.52% | 17.84% | 44.71% | 57.99% |
| CL + KL (Li et al., 2024b) | 27.51% | 56.51% | 69.19% | 23.20% | 50.12% | 62.82% |
| CL + ICL (Yang et al., 2024) | 24.55% | 52.06% | 64.50% | 19.87% | 47.97% | 61.24% |
| CL - TE1 | 30.48% | 62.52% | 74.05% | 24.42% | 54.25% | 68.39% |
| CL - TE2 | 31.80% | 61.37% | 72.90% | **25.19%** | 54.66% | **68.70%** |
| CL - TE1 - TE2 | 32.29% | 62.36% | **75.29%** | 24.50% | 54.56% | 68.14% |
| All Loss Function | **34.76%** | **63.43%** | 74.14% | 24.50% | **55.14%** | 68.29% |

TE with $\gamma > 0$ yields substantial gains in Recall@k, confirming that TE contributes complementary information beyond the standard loss terms. For example, increasing $\gamma$ from 0 to 7.5 improves I2T Recall@1 by over 5% (from 29.00% to 34.10%) and T2I Recall@1 by nearly 5% (from 22.59% to 27.10%). Interestingly, moderate $\gamma$ values (5–7.5) provide the strongest improvements, while excessively large weights (e.g., $\gamma = 10$) slightly degrade performance, likely due to over-regularization. These results highlight that TE not only enhances distillation but also allows for task-specific tuning of the retrieval objectives.

Table 4: Comparison of zero-shot retrieval performance (Recall@k) of student RN34 with teacher RN50 on Flickr8k.

| $\alpha$ | $\beta$ | $\delta$ | $\gamma$ | I2T R@1 | I2T R@5 | I2T R@10 | T2I R@1 | T2I R@5 | T2I R@10 |
|---|---|---|---|---|---|---|---|---|---|
| 1 | 100 | 1 | 0 | 29.00% | 55.60% | 69.85% | 22.59% | 51.07% | 64.66% |
| 1 | 100 | 1 | 1 | 33.20% | 62.19% | 73.72% | 26.85% | 55.32% | 68.34% |
| 1 | 100 | 1 | 5 | 34.76% | 63.43% | 74.14% | 24.50% | 55.14% | 68.29% |
| 1 | 100 | 1 | 7.5 | 34.10% | **63.92%** | **75.29%** | **27.10%** | **55.45%** | **69.03%** |
| 1 | 100 | 1 | 10 | 33.77% | 63.92% | 73.81% | 25.06% | 54.40% | 67.22% |
| 5 | 100 | 1 | 7.5 | 31.88% | 62.52% | 72.82% | 23.16% | 52.09% | 66.00% |
| 1 | 50 | 1 | 7.5 | **34.93%** | 63.59% | 74.88% | 25.12% | 54.79% | 68.11% |
| 1 | 50 | 5 | 7.5 | 31.38% | 61.20% | 74.55% | 25.47% | 55.45% | 68.34% |

### G.3 TEACHER: ViT-B/16, STUDENT: RESNET-34

CLIP ViT-B/16 is a dual-encoder vision-language model (Radford et al., 2021), consisting of a Vision Transformer (ViT-B/16) (Dosovitskiy et al., 2020) as the image encoder and a 12-layer Transformer as the text encoder. The image encoder processes $224 \times 224$ images using $16 \times 16$ patches with a hidden dimension of 768, while the text encoder operates on tokenized text sequences with a hidden dimension of 512. Together, the model has approximately 151 million parameters, with 86M in the image encoder and 63M in the text encoder.

The loss function incorporates weighting factors $\alpha = 1.0$, $\beta = 100$, $\delta = 1.0$, $\gamma = 5.0$, along with a temperature parameter $\tau = 0.07$. These weighting parameters were chosen based on the relative contribution of each loss term to the total loss during training. For the experiments on the MSCOCO dataset, due to the large scale of both the model and the dataset, we trained the student model for 6 epochs. Each experiment (i.e., each row in Table 5) required approximately 10 hours on a Google Colab T4 GPU with high-RAM. For the Flickr8k experiments, we used a Google Colab A100 GPU and trained for 10 epochs. Given the smaller dataset size, each experiment (i.e., each row in Table 7) took around 30 minutes to complete.

Our experiments (Tables 5 and 7) show that maximizing the information flow from teacher to student via TE delivers the single largest boost among all losses. Loss functions with TE leading to the 3–4 percentage point (pp) gains on MSCOCO and the 8–12pp gains on the low-resource Flickr8k benchmarks. These results establish TE as a principled and highly effective regularizer for cross-modal knowledge distillation.

Table 5: Comparison of zero-shot retrieval performance (Recall@k) of student RN34 with teacher ViT-B/16 on **MSCOCO** in VLM distillation with different loss functions. All Loss Function: CL + KL + MSE + ICL - TE1 - TE2.

| Model and Loss Function | I2T Retrieval (R) | | | T2I Retrieval (R) | | |
|---|---|---|---|---|---|---|
| | R@1 | R@5 | R@10 | R@1 | R@5 | R@10 |
| **Teacher Model (ViT-B/16)** | 17.80% | 34.10% | 42.44% | 14.71% | 29.87% | 38.26% |
| **Student Models (RN34)** | | | | | | |
| CL Only (Oord et al., 2018) | 4.66% | 14.10% | 21.28% | 3.78% | 11.95% | 18.40% |
| CL + MSE (Yang et al., 2024) | 4.55% | 14.27% | 21.36% | 3.79% | 11.99% | 18.44% |
| CL + KL (Li et al., 2024b) | 4.70% | 14.46% | 22.21% | 4.58% | 14.15% | 21.32% |
| CL - TE1 | 7.24% | 19.88% | 28.55% | 5.68% | 16.22% | 23.71% |
| CL - TE2 | 7.02% | 20.26% | 29.46% | 5.83% | 16.54% | 24.27% |
| CL - TE1 - TE2 | 7.44% | 20.24% | 29.01% | 5.78% | 16.35% | 23.90% |
| ALL Loss Function | **7.87%** | **21.47%** | **30.74%** | **5.98%** | **17.21%** | **24.96%** |

Table 6 shows the hyperparameter sensitivities for different choices of $\alpha$, $\beta$, $\delta$, and $\gamma$ in the loss function (18). The parameter $\gamma$ controls the strength of the transfer entropy (TE) term. When $\gamma = 0$, corresponding to the absence of TE, the student model performs poorly, with Recall@1 scores of only 5.81% for image-to-text (I2T) and 5.60% for text-to-image (T2I). Introducing a nonzero $\gamma$ immediately leads to substantial improvements across all metrics. For example, setting $\gamma = 1$ raises I2T Recall@1 to 7.32% and T2I Recall@1 to 6.75%, showing that even a small weighting of TE contributes significantly to knowledge transfer.

Table 6: Comparison of zero-shot retrieval performance (Recall@k) of student RN34 with teacher ViT-B/16 on MSCOCO ($\alpha = 1$, $\beta = 50$, $\delta = 1$).

| $\gamma$ | I2T R@1 | I2T R@5 | I2T R@10 | T2I R@1 | T2I R@5 | T2I R@10 |
|---|---|---|---|---|---|---|
| 0 | 5.81% | 17.30% | 25.89% | 5.60% | 16.64% | 24.46% |
| 1 | 7.32% | 20.41% | 29.58% | 6.75% | 18.74% | 26.91% |
| 2.5 | 7.68% | 21.16% | 30.20% | 6.70% | 18.74% | 26.78% |
| 5 | 7.87% | 21.47% | 30.74% | 5.98% | 17.21% | 24.96% |
| 7.5 | 7.90% | 21.27% | 30.22% | 5.92% | 16.69% | 24.42% |
| 10 | 7.49% | 20.56% | 29.51% | 5.47% | 16.01% | 23.38% |

Performance continues to improve as $\gamma$ increases up to 5, with the best I2T results observed at $\gamma = 7.5$ (7.90% Recall@1, 30.22% Recall@10). However, the T2I results peak earlier, with $\gamma = 1$ providing the strongest Recall@1 and Recall@5 values, while larger $\gamma$ values cause a mild decline. This indicates that while TE is generally beneficial, excessively weighting it can distort the loss balance and harm retrieval performance on certain tasks. Overall, these results demonstrate two key points: (i) TE is a crucial component of the loss, consistently lifting performance above the no-TE baseline, and (ii) the optimal $\gamma$ value is task-dependent, suggesting that moderate TE weighting is sufficient to maximize the gains from information-theoretic regularization.

### G.4 TEACHER: RN50, STUDENT MODEL: RN18

In addition to using RN34 as the student model, we also conduct experiments with RN18 as the student image encoder. The RN18 architecture is a more compact variant, containing approximately 11.7 million parameters (He et al., 2016). Similar to RN34, the final fully connected layer is modified to output 1024-dimensional features, keeping the overall parameter count stable. Given that the text

Table 7: Zero-shot retrieval performance (Recall@k) on **Flickr8k**. The student model (RN34) is distilled from the teacher model (**ViT-B/16**). All Loss Function: CL + MSE + KL + ICL - TE1 - TE2.

| Model and Loss Function | I2T Retrieval (R) | | | T2I Retrieval (R) | | |
|---|---|---|---|---|---|---|
| | R@1 | R@5 | R@10 | R@1 | R@5 | R@10 |
| **Teacher Model (ViT-B/16)** | 57.41% | 82.70% | 90.61% | 55.02% | 81.63% | 87.64% |
| **Student Models (RN34)** | | | | | | |
| CL | 21.09% | 46.95% | 59.47% | 17.53% | 42.59% | 55.26% |
| CL + KL | 24.38% | 51.32% | 63.84% | 19.59% | 46.97% | 60.03% |
| CL + MSE | 21.17% | 46.46% | 58.98% | 16.26% | 42.83% | 55.37% |
| CL + ICL | 26.44% | 52.14% | 65.32% | 20.44% | 47.69% | 61.61% |
| CL - TE1 | 28.91% | 57.17% | 69.19% | 22.98% | 50.69% | 63.79% |
| CL - TE2 | 30.07% | 57.41% | 68.45% | 23.67% | 52.04% | 65.44% |
| CL - TE1 - TE2 | 28.42% | 58.90% | 70.02% | 22.59% | 51.10% | 64.79% |
| All Loss Function | **33.28%** | **64.33%** | **73.97%** | **26.36%** | **56.18%** | **69.64%** |

encoder remains unchanged, the total number of parameters for the RN18-based student model is approximately 45-50 million. This reduction in model size compared to the RN34-based student allows for a more lightweight design while still leveraging the benefits of contrastive learning and effective knowledge transfer from the teacher model.

We applied the same loss components and hyperparameter settings as in Section 5: $\alpha = 1.0$, $\beta = 50$, $\delta = 1.0$, $\gamma = 1.0$, and a temperature parameter $\tau = 0.05$. Figure 4 presents the training losses and TE rewards over epochs for various configurations. Compared to RN34, RN18 exhibits a similar trend where the total training loss steadily decreases, and TE rewards increase over epochs, indicating effective optimization and knowledge transfer. However, due to the smaller capacity of RN18, the absolute TE rewards remain slightly lower than those observed for RN34, suggesting a less expressive feature alignment between teacher and student. Furthermore, the KL loss and MSE components show even less significant reductions over training epochs, likely due to the more limited representational capacity of RN18. This highlights that while TE-based regularization remains effective in guiding knowledge distillation, the overall learning dynamics are constrained by the smaller network size, making RN34 a more effective student model in terms of retaining structured alignment with the teacher.

We used Google Colab Pro with a T4 GPU and High-RAM for training and evaluating RN18. Due to its significantly fewer parameters compared to RN34, the student model RN18 required less training time. We trained it for 10 epochs in each loss function combination scenario, with the training and evaluation process taking approximately 11 hours per experimental setup.

We summarize the zero-shot retrieval performance for the trained RN18 student model in Table 8. Similar observations we can make that the experiment with loss function (Contrastive - TE1 -TE2) achieved the best performance for Image-to-Text retrieval, while the experiment with loss function (Contrastive + KL + MSE + ICL - TE1 - TE2) achieved the best performance in Text-to-Image Retrieval. Comparing Table 8 with Table 1, the results indicate that while RN18 achieves competitive performance across different loss function combinations, it underperforms compared to RN34 for all loss configurations, with RN34 consistently yielding higher Recall@k values. However, the best-performing RN18 model (Contrastive - TE1 - TE2) achieves Recall@1 of 6.65% for image-to-text retrieval, which is not far behind RN34's highest Recall@1 values of 8.24% under the same loss formulation. This suggests that while RN18 is a lighter-weight alternative, RN34 remains a better choice for preserving retrieval performance during distillation. The trade-off between model complexity and retrieval accuracy highlights the importance of selecting an appropriate student architecture based on deployment constraints and performance requirements.

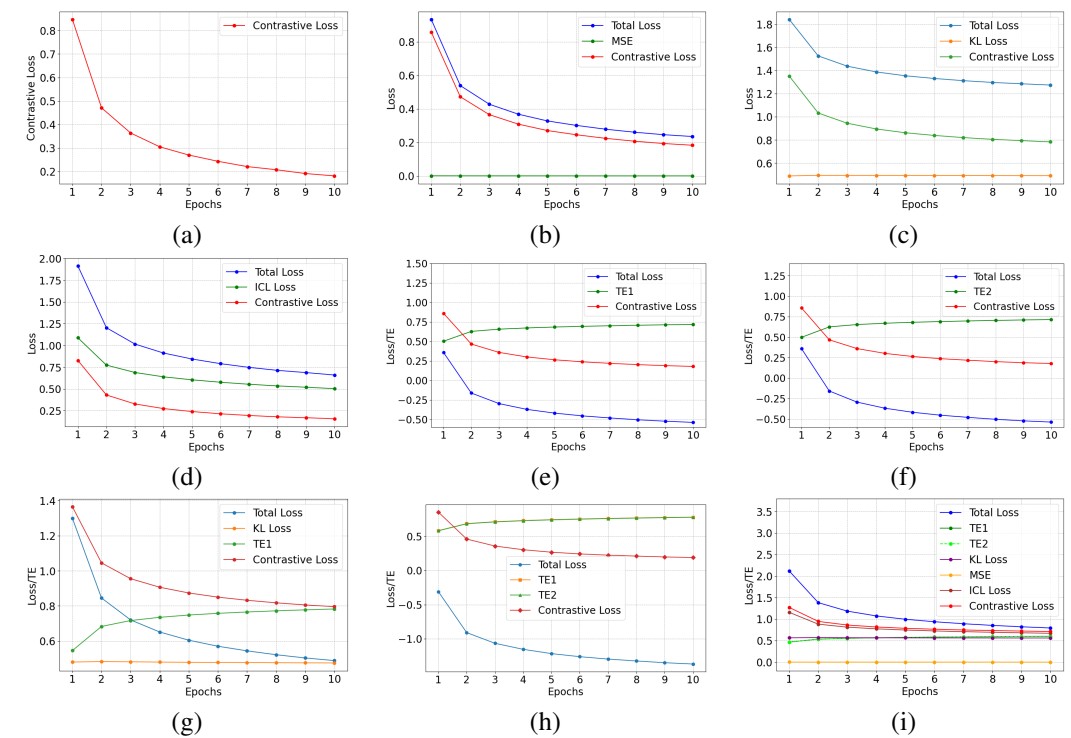

Figure 4: The training losses and TE for different loss functions in the training of Student Model RN18. (a) Contrastive only, (b) Contrastive + MSE, (c) Contrastive + KL, (d) Contrastive + ICL, (e) Contrastive - TE1, (f) Contrastive - TE2, (g) Contrastive + KL - TE1, (h) Contrastive - TE1 - TE2, (i) Contrastive + KL + ICL + MSE - TE1 - TE2.

Table 8: Comparison of zero-shot retrieval performance (Recall@k) of student RN18 with teacher RN50 for different loss function combinations in VLM distillation using MSCOCO. All Loss Function: CL + KL + MSE + ICL - TE1 - TE2.

| Model and Loss Function | I2T Retrieval (R) | | | T2I Retrieval (R) | | |
|---|---|---|---|---|---|---|
| | R@1 | R@5 | R@10 | R@1 | R@5 | R@10 |
| **Teacher Model (RN50)** | 15.27% | 30.73% | 39.05% | 11.68% | 25.52% | 33.50% |
| **Student Models (RN18)** | | | | | | |
| CL Only (Oord et al., 2018) | 4.38% | 13.28% | 20.40% | 3.39% | 11.07% | 17.22% |
| CL + MSE (Yang et al., 2024) | 4.27% | 13.29% | 20.15% | 3.47% | 11.17% | 17.28% |
| CL + KL (Li et al., 2024b) | 4.89% | 15.23% | 22.90% | 4.58% | 13.99% | 21.05% |
| CL + ICL (Yang et al., 2024) | 5.39% | 15.48% | 22.95% | 4.32% | 13.23% | 19.96% |
| CL - TE1 | 5.48% | 16.43% | 24.59% | 4.60% | 13.86% | 20.80% |
| CL - TE2 | 5.57% | 16.67% | 24.78% | 4.67% | 14.08% | 20.97% |
| CL - TE1 - TE2 | **6.65%** | **18.75%** | **27.33%** | 5.18% | 15.09% | 22.35% |
| CL + KL - TE1 | 6.49% | 18.37% | 26.83% | 5.18% | 15.17% | 22.52% |
| All Loss Function | 6.52% | 18.60% | 27.16% | **5.79%** | **16.78%** | **24.47%** |

### G.5 TEACHER: RN50, STUDENT: RN34, APPLICATION IN CLASSIFICATION

We have evaluated our TE-based distillation on Food-101 (Bossard et al., 2014), a challenging benchmark dataset for large-scale food recognition. Food-101 contains 101 categories with a total of 101,000 images, split into 75,750 images for training and 25,250 images for testing. This dataset is particularly suitable for evaluating knowledge transfer since it combines significant intra-

class variation with a large number of categories, which makes direct zero-shot transfer difficult for a smaller-capacity student network.

In our setup, the teacher is a ResNet-50 (RN50) model, and the student is a smaller ResNet-34 (RN34). Importantly, during distillation, the student is trained without direct access to the ground truth labels. Instead, it learns only from the outputs of the teacher, thereby relying entirely on the transferred information. This design allows us to directly measure the effectiveness of the proposed TE-based framework in capturing and transferring generalizable knowledge from teacher to student.

Table 9 summarizes the zero-shot classification accuracy of the student RN34 under different weightings of the loss components (cf. Eq. 18), alongside the teacher RN50 baseline. Several key observations emerge. First, the naive baseline where $\gamma = 0$ (i.e., without TE) performs better than the teacher in terms of Top-1 accuracy but slightly underperforms in Top-5 accuracy. Second, once TE is introduced ($\gamma > 0$), we observe consistent improvements across both Top-1 and Top-5 accuracy. For instance, setting $\gamma = 2.5$ increases the student's Top-1 accuracy to 82.46% and Top-5 accuracy to 96.23%, surpassing the teacher by significant margins. Larger $\gamma$ values generally sustain these gains, with $\gamma = 7.5$ yielding the best Top-5 performance (96.62%), and an alternative setting with $\alpha = 5$ and $\gamma = 2.5$ providing the overall best Top-1 accuracy (82.91%). These trends suggest that TE contributes complementary signal during distillation that is not fully captured by conventional loss terms. Each experiment (each row) in Table 9 takes around 45 minutes using Colab with GPU A100.

Table 9: Zero-shot classification accuracy (%) of student RN34 and teacher RN50 on Food-101.

| $\alpha$ | $\beta$ | $\delta$ | $\gamma$ | Top-1 Acc. | Top-5 Acc. |
|---|---|---|---|---|---|
| 1 | 50 | 1 | 0 | 80.23% | 95.22% |
| 1 | 50 | 1 | 2.5 | 82.46% | 96.23% |
| 1 | 50 | 1 | 5 | 82.37% | 96.44% |
| 1 | 50 | 1 | 7.5 | 82.27% | **96.62%** |
| 1 | 50 | 1 | 10 | 82.01% | 96.38% |
| 5 | 50 | 1 | 2.5 | **82.91%** | 96.47% |
| 1 | 100 | 1 | 2.5 | 82.54% | 96.10% |
| 1 | 50 | 5 | 2.5 | 81.07% | 95.30% |
| Teacher | - | - | - | 79.80% | 96.17% |

Overall, our results demonstrate that the student RN34, despite its smaller capacity, is able to not only match but even surpass the teacher RN50 under several configurations. This improvement cannot be attributed to overfitting, since no ground truth labels are used during distillation, but instead highlights the effectiveness of TE-based distillation in transferring structured, generalizable information. This experiment thus provides strong evidence that TE is a valuable component for enhancing knowledge transfer in classification tasks.

# H   USAGE OF LARGE LANGUAGE MODELS

In preparing this paper, we used ChatGPT 5 to assist with both writing and experimentation. Specifically, it supported text refinement tasks such as grammar correction, spelling, word choice, and stylistic polishing. In addition, it facilitated our experiments by helping to identify and resolve bugs in Python code implementations.

