# OpenReview forum: "TE-VLM: Transfer Entropy for Vision Language Model Distillation"
_ICLR.cc/2026/Conference — ICLR 2026 Conference Withdrawn Submission_

### Official Review · Reviewer_F8hm · 2025-10-30

**Soundness:** 2
**Presentation:** 3
**Contribution:** 1
**Rating:** 2
**Confidence:** 5

**Summary:**

This paper proposes Transfer Entropy (TE) as a regularization term to enhance information flow from the teacher to the student model. TE quantifies the directional dependency between teacher and student embeddings, encouraging the student model to effectively capture structural knowledge from the teacher. Experiments on MSCOCO 2014 and Flickr8k datasets show the proposed method achieves better performance than previous CLIP-KD.

**Strengths:**

1.	This paper is well-written and easy to understand.
2.	The experimental details are complete. The code is provided, making this paper easy to follow.
3.	Experiments on MSCOCO 2014 and Flickr8k datasets show the proposed method achieves better performance than previous CLIP-KD.

**Weaknesses:**

1.	The core idea of the proposed method is to minimize the cosine similarity of linear inter-sample differences between student and teacher. Intuitively, it is equivalent to minimizing the cosine similarity of matched student and teacher features. Therefore, this method does not have a reasonable motivation. And I think this simple idea does not bring much novelty to the VLM distillation field.
2.	The experiments are not convincing. The experimental datasets are small compared to popular image-text datasets, such as LAION-400M, YFCC-15M, or CC-12M. The proposed method should be applied to large-scale multi-modal pretraining to demonstrate its effectiveness on downstream retrieval or classification tasks like ImageNet. Moreover, the student networks are only focused on CNNs. ViT-based networks should be further studied.
3.	The compared methods are not enough. This paper only compares CLIP-KD losses. However, many related CLIP distillation methods [1, 2] are lacking for comparison.

Reference:

[1] Wu K, Peng H, Zhou Z, et al. Tinyclip: Clip distillation via affinity mimicking and weight inheritance[C]//Proceedings of the IEEE/CVF International Conference on Computer Vision. 2023: 21970-21980.

[2] Yang K, Gu T, An X, et al. Clip-cid: Efficient clip distillation via cluster-instance discrimination[C]//Proceedings of the AAAI Conference on Artificial Intelligence. 2025, 39(20): 21974-21982.

**Questions:**

Please refer to the experimental demands in the weakness section.

---

### Official Review · Reviewer_vjZX · 2025-10-31

**Soundness:** 3
**Presentation:** 2
**Contribution:** 2
**Rating:** 4
**Confidence:** 3

**Summary:**

The paper primarily investigates an issue in current VLM knowledge distillation: the lack of explicit quantification of directional information flow from the teacher to the student model, which results in a loss of sequential and structural information. To address this, the paper proposes the TE-VLM method, which uses transfer entropy as a regularization term. Because precise TE is difficult to compute in VLMs, the authors use two computationally efficient TE surrogates, theoretically justify their approximations, and demonstrate the method's effectiveness through experiments.

**Strengths:**

1.The paper provides a detailed analysis of the limitations of current VLM knowledge distillation methods and proposes TE to address this issue, offering a novel perspective to the community.

2.The paper designs efficient TE surrogates and demonstrates their computational feasibility, effectively addressing the intractability of TE in high-dimensional spaces.

3.The experimental comparisons are comprehensive, and the performance improvements are significant, proving the method's effectiveness and generalization ability.

**Weaknesses:**

1.The paper mentions that due to time and budget constraints, the student model was not trained for many epochs. On a dataset of MSCOCO's scale, 10 epochs may not be sufficient for full convergence.

2.Some of the paper's assumptions might be too strong. For example, in Section 4.2.1, when approximating TE by calculating differences between consecutive embeddings, it is assumed that batch order approximates temporal order. This result seems more indicative of the teacher and student embedding spaces having similar smoothness within the batch.

3.The paper's structure has some issues. For instance, Figure 3 is cited in the main body but is located in the appendix. I believe all content in the main paper should ideally be self-contained.

4.The paper lacks a concrete analysis of computational overhead. Although Appendix E provides a theoretical computational complexity analysis of the TE surrogates, the actual (wall-clock time) training overhead added by these TE surrogates (TE1 and TE2) in practice still needs to be reported.

**Questions:**

1.In Table 1, the result for "All Loss Function" being worse than "CL-TE1-TE2" is explained as complementing traditional distillation objectives. However, the absence of a result for "CL+KL+MSE+ICL" might make this conclusion less convincing. I believe it is necessary to add this result.

2.Your proposed TE surrogates (TE1/TE2) compute the cosine similarity of embedding differences between randomly adjacent pairs (i, i+1). Formally, this seems like a highly sparse version of Relational Knowledge Distillation (RKD), as RKD typically considers the relationships between all pairs (i, j) in the batch. Have you compared the effect of extending your TE surrogate to all (i, j) pairs within the batch?

3.Table 2 demonstrates the importance of the TE weight $\gamma$. However, the TE regularization is added to a contrastive loss framework, which is highly sensitive to the temperature $\tau$. Have you investigated the interaction between $\gamma$ and $\tau$? For example, would a smaller $\tau$ change the optimal $\gamma$ value?

---

### Official Review · Reviewer_nmpr · 2025-11-01

**Soundness:** 2
**Presentation:** 3
**Contribution:** 2
**Rating:** 2
**Confidence:** 5

**Summary:**

The paper proposes using Transfer Entropy (TE), an information-theoretic quantity, to improve the knowledge distillation process for CLIP-style (vision-language) models.

**Strengths:**

1. The application of transfer entropy to knowledge distillation is a novel concept. The proposed method's simplicity of implementation is a significant advantage, making it practically attractive.
2. The paper introduces novel approximations for TE that effectively address the high computational complexity of its standard formulation.

**Weaknesses:**

1. The empirical evaluation is very limited, focusing solely on retrieval tasks. Standard benchmarks for CLIP distillation typically include both zero-shot classification and retrieval. I strongly recommend the authors evaluate their method on a broader range of tasks, such as the Image Classification in the Wild benchmark from ELEVATER [1] along with zero-shot classification on ImageNet, IN-V2, IN-A, and IN-A.
2. The training dataset is not clearly specified. The authors should clarify if the MS-COCO 2014 training subset was used, or provide precise details on the dataset.
3. The reported performance metrics (e.g., R@1 in single digits) appear surprisingly low. The authors should investigate and provide an explanation for these results, as they seem to underperform typical baselines.
4. The results lack consistency across the tasks presented in Table 1. For instance, the 'CL-TE1-TE2' configuration excels at I2T retrieval, while the "all loss" combination performs best for T2I retrieval. This variance requires further analysis and discussion.

[1] Li, Chunyuan, et al. "Elevater: A benchmark and toolkit for evaluating language-augmented visual models." Advances in Neural Information Processing Systems 35 (2022): 9287-9301.

**Questions:**

1. To confirm my understanding of the algorithm: At a given time step 't', the teacher model generates $(V_t^T, S_t^T)$ and the student model generates $(V_t^S, S_t^S)$. The optimization step then aims to maximize the TE, such that the next state of the student, $(V_{t+1}^S, S_{t+1}^S)$, contains more information about the current teacher state, $(V_t^T, S_t^T)$, than the current student state $(V_t^S, S_t^S)$ did. Is this interpretation correct?
2. [MINOR] The paper focuses specifically on CLIP models. However, the proposed TE-based distillation method appears general. Could the authors comment on its applicability to other knowledge distillation tasks beyond vision-language models?

---

### Official Review · Reviewer_XdUs · 2025-11-03

**Soundness:** 3
**Presentation:** 3
**Contribution:** 2
**Rating:** 4
**Confidence:** 2

**Summary:**

The paper proposes a new distillation framework that uses Transfer Entropy (TE) to explicitly model the directional flow of information from a large teacher VLM (like CLIP) to a smaller student. It introduces two cosine-similarity–based approximations of TE to make this measure computationally feasible in high-dimensional multimodal embeddings. By adding TE as a regularization (reward) term to standard contrastive, KL, MSE, and ICL losses, the method improves the student’s ability to mimic the teacher’s representational dynamics, achieving gains in image-text retrieval performance across MSCOCO and Flickr8k datasets.

**Strengths:**

1. The paper introduces Transfer Entropy as a novel, theoretically grounded measure of directional knowledge flow for VLM distillation.

2. The paper derives a first-order TE–Jacobian relation, offering a principled and interpretable foundation for the proposed regularization.

3. The paper proposes two efficient cosine-similarity–based TE approximations that make TE computation feasible in high-dimensional multimodal spaces.

**Weaknesses:**

* **Limited empirical scope:** Experiments are confined to smaller datasets (MSCOCO, Flickr8k) and moderate-scale models, leaving open questions about scalability to large VLMs or real-world tasks.
* **Significant teacher–student gap:** Despite improvements, the student’s performance still lags notably behind the teacher, indicating limited knowledge transfer efficiency.
* **Lack of interpretability analysis:** The paper does not visualize or qualitatively analyze what kinds of information TE actually transfers between modalities.
* **Shallow ablation on loss interactions:** The interplay among TE, KL, MSE, and ICL terms is discussed but not systematically disentangled or analyzed.
* **Efficiency trade-offs unquantified:** The additional computational cost of TE-based regularization versus standard distillation is not clearly reported.
* **Incomplete theoretical validation:** The proposed TE–Jacobian approximation is not empirically compared with mutual information–based baselines to confirm its theoretical soundness.

**Questions:**

* **Quantitative efficiency:** How much additional computational cost (training time or GPU memory) does the TE regularization introduce compared to standard distillation?
* **Scalability:** Have you tested the method with larger teacher–student pairs (e.g., ViT-L → ViT-B) or on larger datasets (e.g., Flickr30k, Conceptual Captions) to verify scalability?
* **Interpretability of TE:** Can you provide qualitative or visualization evidence (e.g., feature trajectory or embedding alignment plots) showing what kind of information TE captures or transfers?
* **Theoretical grounding:** How well does the proposed TE–Jacobian approximation correlate with true transfer entropy or mutual information estimates in lower-dimensional settings?
* **Remaining performance gap:** What do you hypothesize as the main cause of the residual teacher–student performance gap, and how might TE-based distillation be extended to close it further?

---

### Note · Authors · 2025-11-14

I have read and agree with the venue's withdrawal policy on behalf of myself and my co-authors.